# Design and Analysis of a PDLC-Based Reconfigurable Hilbert Fractal Antenna for Large and Fine THz Frequency Tuning

**DOI:** 10.3390/mi13060964

**Published:** 2022-06-18

**Authors:** Prabir Garu, Wei-Chih Wang

**Affiliations:** 1Institute of Nano Engineering and Micro Systems, National Tsing Hua University, Hsinchu City 300, Taiwan; prabir.garu92@gmail.com; 2Department of Power Mechanical Engineering, National Tsing Hua University, Hsinchu City 300, Taiwan; 3Department of Mechanical Engineering, University of Washington, Seattle, WA 98195, USA; 4Department of Electrical Engineering, University of Washington, Seattle, WA 98195, USA

**Keywords:** Hilbert fractal, reconfigurable antenna, frequency tuning, polymer dispersion liquid crystal

## Abstract

The proposed reconfigurable radiating antenna design is based on the integration of a reconfigurable fractal antenna and electro-optic substrate material. This antenna can be adjusted to achieve either re-configurability or tunability in the desired frequency range for wireless systems. The electromagnetic characteristics of the fractal antenna are manipulated at both the level of fractal geometry, electrical length and dielectric substrate. The designed antenna features multiband responses, in which the geometry and length change create a large frequency shift and the dielectric change using polymer dispersed liquid crystal (PDLC) creates fine and/or continuous tuning. The far field and scattering properties of the antenna are analyzed using the Computer Simulation Technology (CST) Microwave Studio Suite. The proposed approach has successfully demonstrated reconfigurable switching for up to four frequency bands between 0.2 and 0.6 THz. The dielectric constant change in the PDLC substrate shows fine and continuous frequency tuning with an 8% maximum frequency shift when operating around 0.54 THz and a high directivity of 7.35 dBi at 0.54 THz and 8.43 dBi at 0.504 THz. The antenna can also realize a peak gain of 4.29 dBi at 0.504 THz in the extraordinary polarization state of PDLC. The designed antenna can be readily integrated in the current communication devices, such as satellites, smart phones, laptops, and other portable electronic devices, due to its compact geometry and IC compatible design. In satellite applications, the proposed antenna can play a significant role in terms of security. The antenna could be extremely useful for satellites that want to keep their information secret; by constantly switching their operating frequency, spy satellites can evade detection and data collection from enemy ears.

## 1. Introduction

Due to the rapid advancements in modern communication, fractal reconfigurable antennas (FRAs) have spurred huge research interests, owing to their various demanding applications in THz. Such antennas are quickly becoming the current field of interest for researchers, thanks to significant advancements in antenna engineering research. In order to provide various functionalities for future wireless communication systems, a reconfigurable antenna with tunable fundamental characteristics is highly preferred. The antenna’s configurability allows it to tune its working frequency, which can operate as a filter to prevent interfering signals. The antenna can adapt to a wide range of environments and applications. Fractal features, such as self-similarity [1], self-affinity [2], and space-filling [3], combined with electrical reconfigurability to dynamically modify multiple antenna parameters, have made FRAs very appealing for multifunctional antenna designs. The majority of FRAs use traditional microstrip patch antennas, which incorporate fractal geometries and reconfigurability as part of the antenna design process. Decades ago, individual antennas were used to achieve different working frequencies, which increased the system’s dimensions. Conversely, ultra-wideband (UWB) antennas can be utilized to attain a wide frequency range (3.1–10.6 GHz) [4]. However, employing UWB antennas has several drawbacks, such as interference and coexistence with other radios. To avoid these problems, the concept of reconfigurability has considerably evolved. The antenna size has been greatly decreased thanks to frequency reconfigurability [5], making them suitable for use in any small handheld device. Instead of using multiple antennas that operate at different frequencies for signal transmission or reception, a frequency reconfigurable antenna is likely the most feasible solution for switching its operation to the desired frequency. There are three other categories of reconfigurable antennas, which include pattern reconfigurable antennas [6,7], polarization reconfigurable antennas [8] and compound reconfigurable antennas [9], based on the operational antenna parameter that is dynamically adjusted. Reconfigurable antennas with varied fractal designs can be used effectively for applications in a wide range of frequency bands. Depending on the number of switches, two or more reconfigurabilities can be implemented at the same time. The cognitive radio system, in which an antenna adjusts to the appropriate frequency based on the available spectrum, represents the near future of wireless communication [10]. Accordingly, reconfigurable antennas, which are utilized in satellite communications and radar systems, are one of the best choices for such adaptive systems. To solve single or dual frequency band operation problems, a multiband antenna can be used where a single antenna can operate at many frequency bands. Applying a fractal shape to antenna design is one technique to make a multiband antenna. Fractal technology allowed for the designing of miniature antennas and integrating multiple bands into a single device. Properties such as space filling, self- similarity, fractional dimensions, infinite complexity, mechanical simplicity and robustness make the fractal antenna unique to attain advantages such as miniaturization, wideband and multiband characteristics with better efficiency. The space filling property is used to reduce the antenna size and self-similarity to achieve the multiband resonant antenna. The number of iterations of these geometries is based on operating wavelengths. Although fractal antennas minimize size and cost, in the case of the communication system, many applications require many different frequency bands. Therefore, a single fractal antenna cannot serve the function of the entire communication system. To improve spectrum sharing and expand operational frequency ranges, one is likely to consider reconfigurable radio frequency (RF) circuits and devices. The current reconfigurable antennas for mobile terminals integrate different active devices for fine tuning their operation frequencies [11,12]. This antenna offers compact size structures and improved gain and radiation patterns compared to conventional antennas. The resonating frequency, operating bandwidth and directivity of such antennas can be manipulated by introducing RF switches [13]. Based on the requirement on the reconfiguration property of the antenna, there are four major types of reconfiguration techniques, electrical, optical, mechanical and material [14,15]. The reconfiguration techniques are presented in Figure 1. Existing reconfiguration solutions for RF switches are based on positive-intrinsic-negative (PIN) diodes [16,17], microelectromechanical systems (MEMS) switches [18], varactor diodes [19], lumped elements [20], tunable materials [21,22], or optical switches [23]. Depending on the ON and OFF conditions, these devices change the structure (length) of the radiating element. PIN diodes are mainly used in switching on and off certain parts of the antenna, and hence achieve frequency reconfigurability. Baruah et al. have proposed an electrically controllable rectangular microstrip patch antenna, which offers frequency reconfigurability using PIN diodes [24]. In the design, PIN diodes are incorporated to provide necessary electrical connectivity between the main radiator and the parasitic patches. The antenna offers frequency reconfigurability over a frequency range of ~1.0 GHz with seven different frequency bands and constant radiation characteristics at all the reconfigured frequencies. PIN diodes present good performances at frequencies up to 20 GHz and are similar to field-effect transistor (FETs) devices, however, with the cost of high power consumption. Shah et al. have demonstrated novel shaped hexa-band frequency-reconfigurable antenna with a very wide tuning band [25]. The proposed antenna operates at two single and dual band modes depending upon the switching states. For the numerical study, lumped components are employed to generate tunable capacitance, which is responsible for frequency reconfigurability. The fabricated antenna has wide tuning capability, which spans from 2.1 to 5.2 GHz. Erdil et al. presented a reconfigurable microstrip patch antenna that is monolithically integrated with RF MEMS capacitors for tuning the resonant frequency [15]. The reconfigurability of the patch antenna is achieved by loading it with a coplanar waveguide (CPW) stub on which variable MEMS capacitors are inserted at periodic intervals. The surface micromachining technique is used to create MEMS capacitors, which consist of a 1 µm thick aluminum structural layer placed on a glass substrate, with a capacitive gap of 1.5 um. Low tuning values in the range of 0–11.9 V are used to electrostatically actuate MEMS capacitors. With 0 to 11.9 V actuation, the antenna resonant frequency can be continuously shifted from 16.05 GHz down to 15.75 GHz. However, the MEMS-based systems have a low tuning speed and need complex integration. A dual-band reconfigurable terahertz patch antenna composed of microstrip and graphene backing cavity has been demonstrated by Dong et al. [26]. Patch resonance is used in the proposed antenna, which is defined by interleaved graphene/Al_2_O_3_ stacks. Through electrostatic gating on the graphene stack, it can be dynamically dual resonance frequency-tuned across a broad range of frequency roughly 1 THz. The direction of the antenna main beam may be steered with a significant variation range by applying different voltages to the graphene stack. These initial findings are particularly promising for future THz applications, especially THz antenna arrays. Radwan et al. demonstrated a tunable THz radiation band antenna, whose resonance frequency and bandwidth can be changed by the applied voltage [27]. The proposed structure offers a wider operational bandwidth and a better match between the photomixer and the antenna. Using an array of split ring resonators (SRR), the bandwidth and radiation properties can be improved. M. Tamagnone et al. developed a reconfigurable adjustable graphene dipole antenna with excellent impedance stability [28]. This approach provides excellent properties in terms of frequency-reconfiguration, and stable impedance and radiation patterns upon tuning across the entire operation band. These findings hold great promise for future graphene-based monolithic integration of the antenna and THz emitter/detector.

Despite the fact that all of these reconfigurable antennas have exhibited a broad range of tunability, the majority of them lack in tuning range or need somewhat sophisticated manufacturing or exotic semiconductor materials. Discrete frequency tuning is another feature of switch-based reconfigurable antennas. Although some simulation results have been reported in the THz range, no reports of actual devices have been demonstrated with tunable frequency and wide band operation. The proposed reconfigurable radiating antenna design is based on the integration of a reconfigurable fractal antenna and electro-optic substrate material. The electromagnetic characteristics of the fractal antenna are manipulated at both the level of fractal geometry, electrical length and dielectric substrate. The designed antenna features multiband responses, in which the geometry and length change create a large frequency shift, whereas dielectric change using polymer dispersed liquid crystal (PDLC) creates fine and/or continuous tuning. The advantages of using PDLC also eliminated the need for LC cells for LC entrainment. This allows a simpler and integrated design because the entire antenna can be fabricated using typical microfabrication. For the future application, we intend to use this antenna design with electronic-based THz devices for a tunable transceiver system. Therefore, a fully integrated system is preferred.

Various iteration stages of the fractal Hilbert curves are shown in Figure 2. It may be observed that geometry at an iteration stage can be obtained by combining four scaled down copies of the previous iteration, connected with three additional line segments. For example, the geometry of Order 2 can be thought of as four copies (nearly halved) of the geometry with Order 1 (arranged in different orientations). The design also utilizes rectangular shape patterns that can easily reconfigured by the switch on and off using the proposed transistor switch system.

To design a proper structure for any antenna, it is desired to calculate the different parameters, and then their dimensions are arranged for the specific bands. For an appropriate design, we have to find the exact dimensions of different elements, such as the width and the length of the structure, top layer, ground layer and many more. The unit cell parameters are shown in Figure 3. The geometric parameters are listed in Table 1. The perturbed fourth order Hilbert curve is used to reduce the losses by shifting the large and smooth frequencies to the higher and lower band. The following approximation was used to obtain the resonance frequency of the structure [29].
(1)$$f_{o}\approx \frac{C_{o}}{2S_{s}\sqrt{\varepsilon_{eff}}}$$
where $C_{o}$ is the speed of light in free space, $\varepsilon_{eff}$ effective permittivity; $S_{s}$ is the length of the perturbed Hilbert curve slot. The $S_{s}$ can be expressed as
(2)$$S_{s}=\left(2^{n}+1\right)l_{s}$$
where n is the curve order and $l_{s}$ is the side length of the curve, as illustrated in Figure 3a,b.

## 2. Results and Discussion

To study the performance of the proposed antenna, the radiating structure is designed, simulated, and analyzed using 3D electromagnetic simulator CST Microwave Studio (MWS) Suite 2021. A waveguide port is used as a source for the excitation of the designed antenna. The reflection coefficient (S11) is evaluated with open-add-space boundary conditions using a transient solver in CST Microwave Studio.

### 2.1. Large Frequency Shift or Tuning

The electromagnetic characteristics of the fractal antenna are manipulated at both the level of fractal geometry and electrical length. Due to its unique repeated fractal pattern, it allows for the design of relatively smaller antenna dimensions for a relatively long wavelength operation. One of the features in our proposed antenna design is to utilize this length reduction in the Hilbert fractal antenna to create a medium to large discrete frequency shift (Figure 4a). As one can observe from Figure 4a, for a large frequency shift, one can simply disconnect a larger section of the antenna’s electrical length (or disconnect a large segment of fractal pattern) in the antenna, hence reducing the overall size of the antenna and generating an upward frequency shift. On the other hand, if more fractal patterns are added or the electrical length is increased, the frequency will decrease due to a longer antenna. For a finer frequency shift, one can simply disconnect fewer fractal patterns or change the electrical length (Figure 4b,c). The effect can either increase the electrical length, which decreases the frequency, or vice versa. A simulation performed on a CST numerical software shows that a 0 to >21% frequency shift can be generated by this reconfigurable Hilbert fractal antenna design when it operates at the 0.3 to 0.4 THz region (Figure 4d). The finesse of the frequency shift and tuning range is highly dependent on the fractal antenna design. For the demonstration of the concept, the Hilbert fractal pattern is utilized and only a slight modification is carried out on the dimension to show the potential of using a reconfigurable fractal antenna. However, many more unique fractal pattern designs will be investigated later using a generic algorithm or wavelet technique to search for the best frequency reconfigurability and performance. PIN diodes, varactors, FETs or other transistors will be used as switches to allow for frequency re-configurability between 0.2 and 0.6 THz [30]. Note that the position of varactors on the antenna will be optimized for obtaining a wide range of frequency tunability. These diodes can be fabricated along with the rectifier or subsequent supporting electronics separately, then anodic bonded together later.

### 2.2. Fine Frequency Shift/Continuous Frequency Tuning

For a fine frequency shift or continuous frequency tuning, we replace the normal passive low dielectric buffer layer underneath the metal antenna with a ferroelectric material, whose permittivity (e) changes when a different electric field is applied. In this case, based on our previous measurement from 0.2 THz to 1.2 THz, the permittivity of PDLC in that frequency band of operation was calculated using the modified Nicolson–Ross–Weir (MNRW) extraction technique. The unit cell antenna based on single segment Hilbert fractal is shown in Figure 5a. As it is different from the fractal pattern where the frequency shift is more discrete and highly depended on its fractal pattern, the voltage induced dielectric constant change in the antenna can create much more continuous and infinitesimal frequency change. An example of this frequency shift due to the permittivity change using PDLC substrate on a Hilbert fractal antenna is shown in Figure 5b. One can clearly observe that the frequency changes proportionally with permittivity. However, the frequency shift (0.504 to 0.540 THz) is due to ε change from 2.93 to 2.51 @ 0.5 THz from extraordinary to ordinary polarization in the PDLC (Figure 5b), which is about the same as the reconfigurable fractal antenna when changes one segment to two segments, as shown earlier in the simulation (Figure 4). Eventually, several larger dielectric constant change ferroelectric or magneto-electric materials (currently developing in the lab) will be developed for the proposed reconfigurable frequency tunable antenna design. The main challenge with integrating the antenna eventually with the oscillator, such as RTD, is the impedance matching. Care must be taken when designing the tunable antenna so an additional tunable impedance matching circuit might be needed to ensure a good energy coupling between the antenna and the rest of the RF components or circuits.

Due to the substrate material chosen currently, the return loss is slightly lower than expected. However, for immediate THz near field application, it will be sufficient. The cross-sectional and 3D perspective view are illustrated in Figure 5c,d, respectively, for the clear observation of PDLC.

### 2.3. Farfield Radiation Pattern

The farfield radiation patterns (3D and polar plot) of the one segment fractal antenna are shown in Figure 6a–d. The measured results of the antenna clearly show that the antenna radiation characteristics change upon the changes in permittivity of the liquid crystal (PDLC) from the ordinary to extraordinary state. In the extraordinary state, the farfield directivity of the antenna is obvious and quite high (highly directive), compared to the ordinary state. The directivity of the antenna in ordinary state is 7.35 dBi at 0.54 THz, and in the extraordinary state, it is 8.43 dBi at 0.504 THz. Under ε state changing, the patterns are slightly deviated (e.g., the beam width, side lobe level, and lobe magnitude), which can be understood from the polar plots (b) and (d) of the antenna. However, the most important point to highlight is that the antenna reconfigurability feature under the state change in the PDLC is well retained. On the other hand, the farfield radiation patterns (3D and polar plot) of the two segment fractal antenna under different physical length variations are shown in Figure 7a–f at the ordinary state of PDLC. The antenna radiation characteristics alter when the electrical length is varied by cutting some parts of the fractal, as observed in the radiation patterns. From Figure 7a,b, it is observed that without the cutting of the fractal arm (no change in electrical length), the directivity of the fractal antenna is 5.9 dBi at 0.558 THz, with a 3dB bandwidth of 89.8 degree and main lobe direction of 3.0 degree. However, the magnitude of the directivity and main love direction dramatically changed when some of the fractal arm was cut into different electrical lengths. When the fractal geometry was cut into small physical lengths (large electrical length), the curve shifted towards the left and the peak directivity changed to 7.22 dBi at 0.555 THz with a 3dB bandwidth of 59.7 degree and main lobe direction of 18 degree, as shown in Figure 7c,d. However, when the fractal geometry was cut into large physical lengths (small electrical length), the curve shifted towards the right and the directivity reached 7.21 dBi at 0.579 THz with a 3dB bandwidth of 74.4 degree and main lobe direction of 30 degree, as depicted in Figure 7e,f. It is worth noting that all the proposed antennas radiate predominantly in the vertical or E plane (θ-plane) for the respective frequency bands. In the operating frequency bands, the antenna radiates very efficiently at the resonant, with a high radiation efficiency of >90%. The farfield directivities of all the fractal antennas at resonant frequencies are summarized in Table 2.

The finite integration approach (FIT) utilized in CST MWS is used to analyze the simulated gain plots of the designed antenna. The far-field gain (3D and polar plot) of the one segment fractal antenna is displayed in Figure 8a–d. The peak gain of the antenna in ordinary state is 1.73 dBi at the resonant frequency 0.54 THz and in the extraordinary state, it is 4.29 dBi at the resonant frequency 0.504 THz. On the other hand, the farfield gain (3D and polar plot) of the two segment fractal antenna under different physical length variations is shown in Figure 9a–f at the ordinary state of PDLC. Initially, a gain of 0.981 dBi was observed at 0.558 THz for the two segment antenna without the cutting of the fractal arm (no change in electrical length), as shown in Figure 9a,b. However, after varying the electrical length value, a significant amount of gain is observed for the two segment fractal antenna. The gain of 2.09 dBi at 0.555 THz is observed when the fractal geometry was cut into small physical lengths (large electrical length). On the other hand, a gain of 2.42 dBi at 0.579 THz is achieved when the fractal geometry was cut into large physical lengths (small electrical length). The farfield gain of all the fractal antennas at resonant frequencies are summarized in Table 2.

## 3. Conclusions

In this work, a frequency reconfigurable Hilbert fractal antenna has been designed and its electromagnetic performance has been numerically studied. The different switching states were responsible for achieving the desired operating characteristics. One of the features of our proposed antenna design is that it makes use of the Hilbert fractal antenna’s length reduction to produce a medium to large discrete frequency shift. The designed proposed fractal antenna can generate a 0 to >17% continuous frequency shift within the operating frequency band. Moreover, the one segment fractal antenna shows a very high gain of 4.29 dBi. In addition, the two segment smaller electrical length fractal antenna achieved a significant amount gain of 2.42 dBi. The proposed antenna has numerous advantages, including its small size, low cost, ease of fabrication, and ease of integration. The designed antenna can be utilized in satellite communications, military applications, and modern communication devices (i.e., laptops and tablets), as well as in RTD devices.

## Figures and Tables

**Figure 1 micromachines-13-00964-f001:**
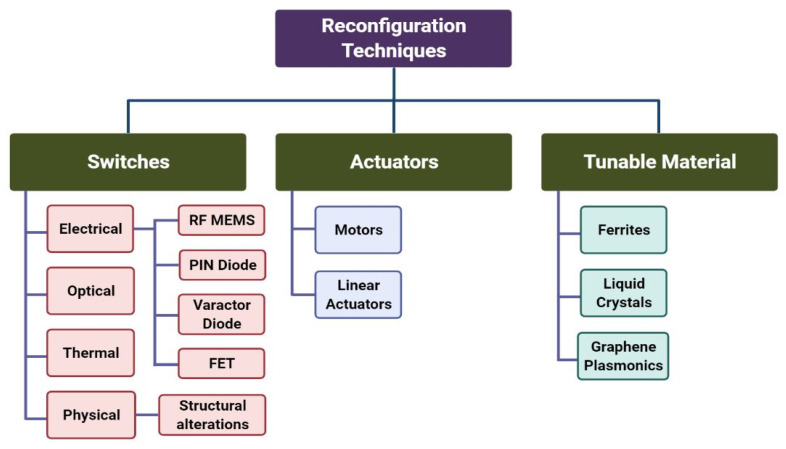
Different techniques of achieving antenna reconfigurability.

**Figure 2 micromachines-13-00964-f002:**
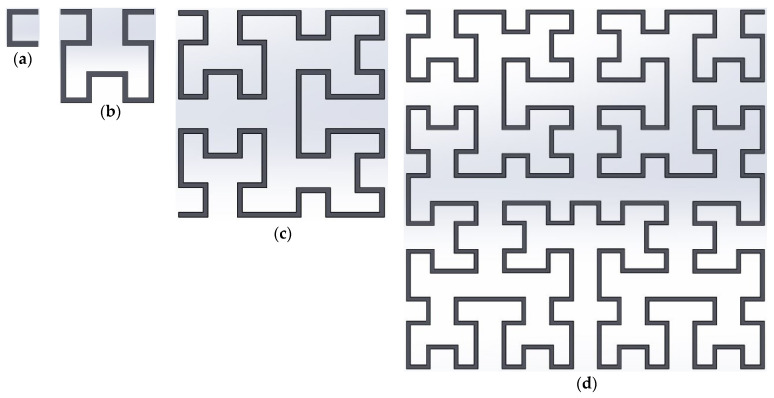
Generation of four iterations of Hilbert curves: (**a**) 1st iteration, (**b**) 2nd iteration, (**c**) 3rd iteration, (**d**) 4th iteration. The additional segments used to connect together copies of the previous iteration curves. First four fractal iterations for the Hilbert curve geometry. This indicates that for a given area, the total length of the line segments increases in near geometric progression, as the iteration order increases.

**Figure 3 micromachines-13-00964-f003:**
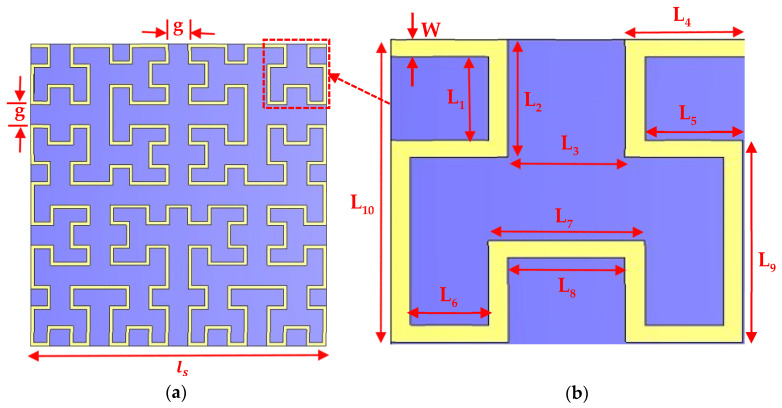
Geometric dimensions of (**a**) one segment unit cell (4th iteration) and (**b**) magnified view of the 2nd iteration of the unit cell antenna.

**Figure 4 micromachines-13-00964-f004:**
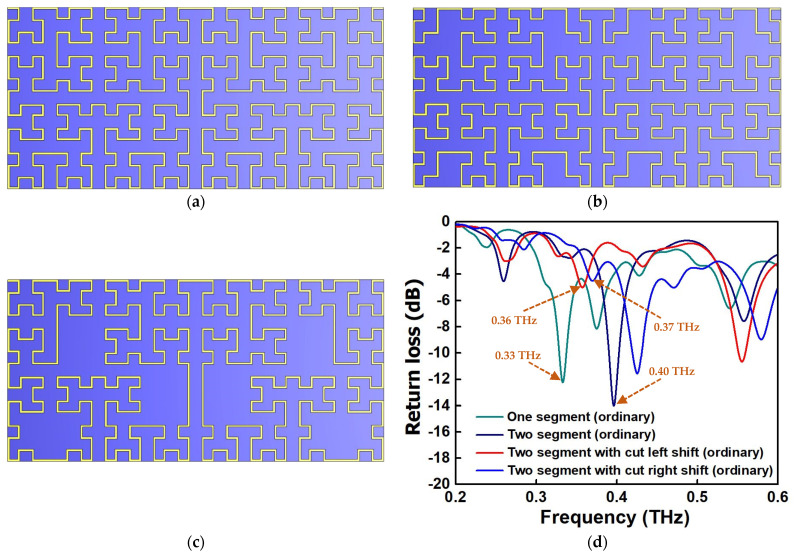
Reconfigurable Hilbert fractal antenna: (**a**) two segments without cut arm, (**b**) two segments with longer electrical length, (**c**) two segments with shorter electrical length and (**d**) large frequency shift by disconnecting or removing some part of the fractal pattern or electrical length. Here, we observe a left ward shift in frequency when the size of the fractal pattern is reducing and when size is increasing, we observe a rightward shift in frequency. This is clearly shown by the switching between one segment and two segment fractal design. Medium range frequency shift can be achieved by removing a smaller portion of the fractal pattern in the design. Here, three arms are cut from each segment, which means that the electrical length increases. A ~0.02 THz downward shifted is observed (leftward shift). With even larger cut and arm length reduction, a rightward frequency shift is observed. Both techniques can be easily implemented by embedding diode-based switches at the junctions in the antenna to turn certain parts of the antenna on and off [30].

**Figure 5 micromachines-13-00964-f005:**
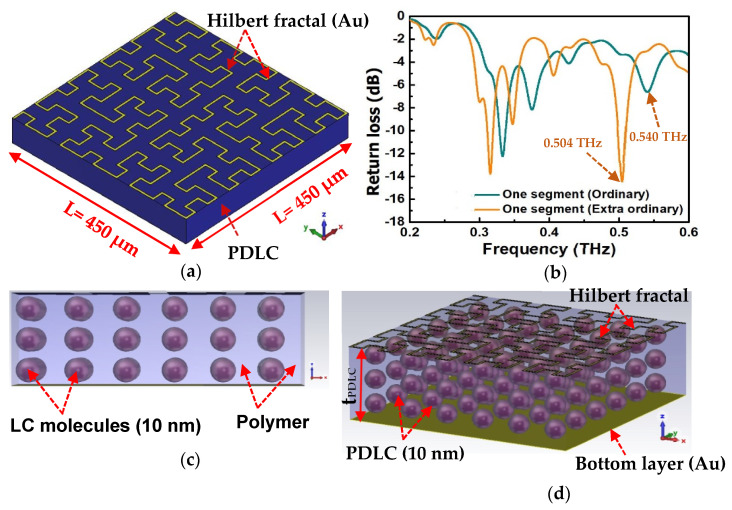
(**a**) One segment unit cell antenna and (**b**) return loss of the proposed antenna from extraordinary to ordinary polarization in the PDLC. Continuous frequency tuning using dielectric changed using ferroelectric material (PDLC). Here, frequency shifts are due to liquid crystal changes from the ordinary to extraordinary state. (**c**) Side view and (**d**) 3D perspective view of the unit cell antenna. The thickness of the PDLC is t_PDLC_ = 50 µm, with a liquid crystal size of 10 nm. The top Hilbert fractal (au) and the bottom metal (Au) thickness is 200 nm.

**Figure 6 micromachines-13-00964-f006:**
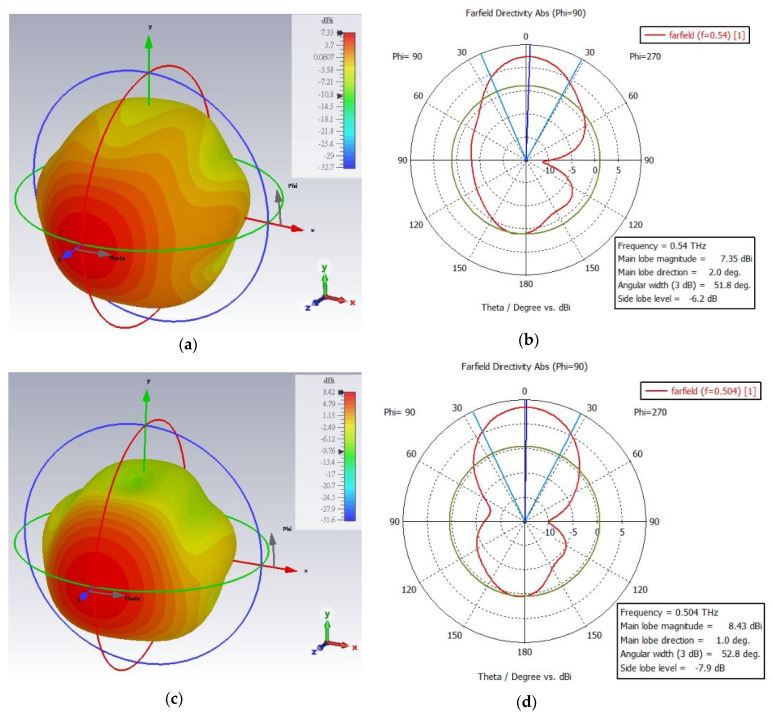
**The** 3D and polar plot of the farfield radiation pattern of the one segment Hilbert fractal reconfigurable antenna from (**a**,**b**) ordinary (at 0.54 THz) to (**c**,**d**) extraordinary (at 0.504 THz) polarization in the PDLC.

**Figure 7 micromachines-13-00964-f007:**
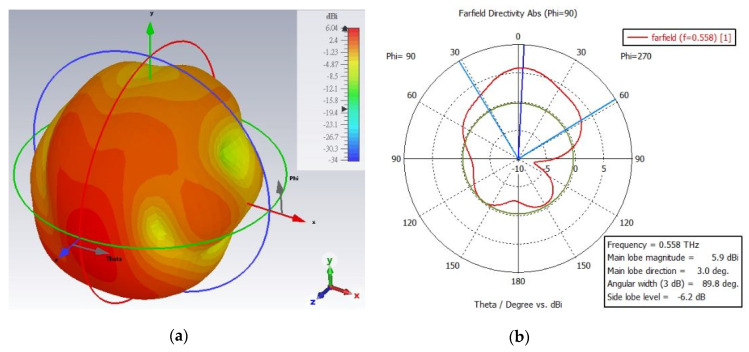
**The** 3D and polar plot of the farfield radiation pattern of different electrical length two segment Hilbert fractal reconfigurable antenna in ordinary polarization in the PDLC. (**a**,**b**) Two segments with no cut (at 0.558 THz), (**c**,**d**) twos segment with cut (large electrical length) left shift (at 0.555 THz), and (**e**,**f**) with cut (small electrical length) right shift (at 0.579 THz) of the fractal geometry.

**Figure 8 micromachines-13-00964-f008:**
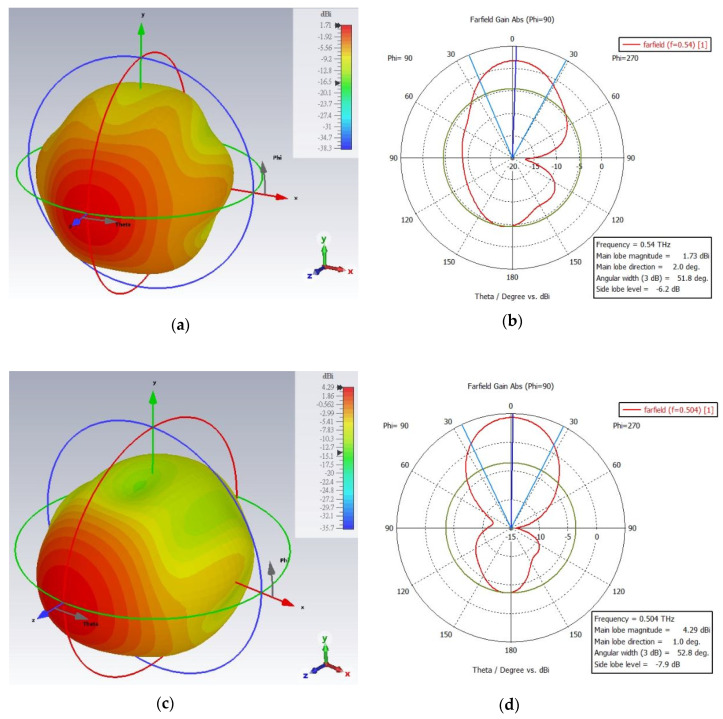
**The** 3D and polar plot of the farfield gain of the one segment Hilbert fractal reconfigurable antenna from (**a**,**b**) ordinary (at 0.54 THz) to (**c**,**d**) extraordinary (at 0.504 THz) polarization in the PDLC.

**Figure 9 micromachines-13-00964-f009:**
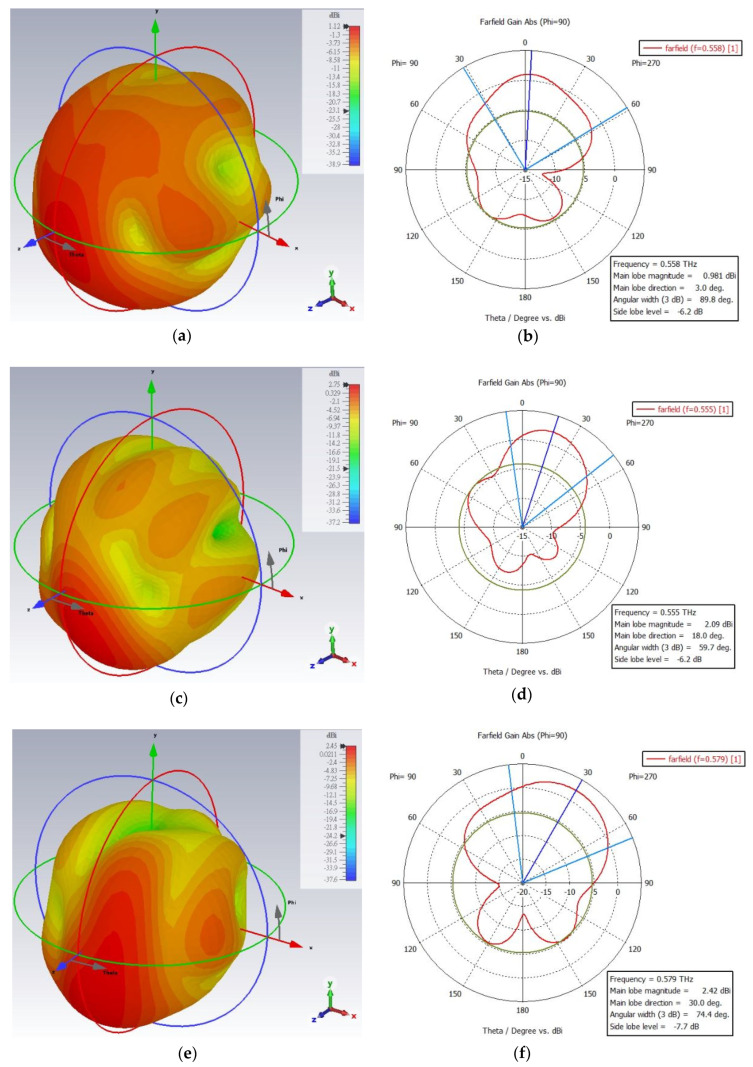
**The** 3D and polar plot of the farfield gain of different electrical length two segment Hilbert fractal reconfigurable antenna in ordinary polarization in the PDLC. (**a**,**b**) Two segments with no cut (at 0.558 THz), (**c**,**d**) two segments with cut (large electrical length) left shift (at 0.555 THz), and (**e**,**f**) with cut (small electrical length) right shift (at 0.579 THz) of the fractal geometry.

**Table 1 micromachines-13-00964-t001:** Optimum values for fractal geometric dimensions of the proposed reconfigurable antenna.

Parameters	L_1_	L_2_	L_3_	L_4_	L_5_	L_6_	L_7_	L_8_	L_9_	L_10_	W	g	$l_{s}$	$S_{s}$
**Dimension (µm)**	25	35	30	30	25	20	40	30	60	90	5	30	450	7650

**Table 2 micromachines-13-00964-t002:** Parametric comparison of different geometries of the fractal antenna at resonant frequencies.

Design Name		Resonant Frequency (f_c_) (THz)	Return Loss (dB)	Gain (dBi)	Directivity (dBi)
One segment	Ordinary state	0.54	−6.61	1.73	7.35
	Extra ordinary state	0.504	−14.5	4.29	8.43
Two segment (without arm cut)		0.558	−8.07	0.981	5.9
Two segment (large electrical length)		0.555	−10.7	2.09	7.22
Two segment (small electrical length)		0.579	−9.4	2.42	7.21

## Data Availability

The data presented in this study are available on request from the corresponding author. The data are not publicly available due to patent pending issues.

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
