# Peer review of "Design and Analysis of a PDLC-Based Reconfigurable Hilbert Fractal Antenna for Large and Fine THz Frequency Tuning"

_micromachines, 2022, doi:10.3390/mi13060964_

Round 1
Reviewer 1 Report
In this paper, a reconfigurable radiating antenna using polymer dispersed liquid crystal was designed. The simulated results are very interesting and helpful. Some comments are as follows:
1, In Figure 1, the authors compared different reconfigurable techniques. For the liquid crystal used in this paper, can you explain the response time between the LC and electrical tuning?
2, In Section 2, please explain to the readers why you choose the Hilbert fractal unit? what's its advantage?
3, Please explain the design goal of this antenna and its application. And in the simulated results (Fig.5b), the return loss and bandwidth seems not very good.
4, In the terahertz frequency band, the loss of the LC should be considered. Please let me know how to set the loss tangent of the LC?
Reviewer 2 Report
In this paper, authors proposed interesting and compact antenna designs based on Hilbert fractal and electro-optic PDLC substrate. The detailed simulation results showed reconfigurable frequency shifting (0.2-0.6THz). The manuscript is written in an organized way (sufficient background and references), and designs were studied with professional simulation tool. I'd suggest publication after addressing some questions/comments below:
1. In abstract, Page 1, Line 21 : a typo of PDLC (PDLD)?
2. In Introduction section, Page 1-2, please illustrate some acronyms : UWB, RF, PIN, FET…
3. Figure 1, page 3, some images (switches, actuators, tunable material) are not readable.
4. Page 6 line 224, how did authors get (calculate) the number 0-17% ? (a CST numerical software shows that a 0 to > 17% frequency shift can be generated by this 224 reconfigurable Hilbert fractal antenna design (Figure 4 (d)).
5. Page 7, ling 293, (An example of this frequency shift 292 due to permittivity change using polymer dispersed liquid crystal (PDLC) substrate on a 293 Hilbert fractal antenna is shown in Figure 5 (b). ) could authors add more details about the PDLC? A) What is the materials – liquid crystal and polymer? B) Or thickness? C) Or LC droplets size? D) Within a glasses sandwiched cell? E) with any voltage applied onto PDLC? F) could authors illustrate more (maybe a side view or 3D sketch) about the cell antenna with PDLC in Figure 5(a)?
6. Could author explain the motivation of using PDLC? As simple LC cell without polymer (LC display) could also change permittivity? Also, will the scattering in PDLC (index mismatch between polymer and LC) affect the signal (frequency tuning)?
